# Ultrasound Description of Follicular Development in the Louisiana Pinesnake (*Pituophis ruthveni*, Stull 1929)

**DOI:** 10.3390/ani12212983

**Published:** 2022-10-30

**Authors:** Matteo Oliveri, Mark R. Sandfoss, Steven B. Reichling, Melanie M. Richter, Jessica R. Cantrell, Zdenek Knotek, Beth M. Roberts

**Affiliations:** 1Faculty of Veterinary Medicine, Teaching Veterinary Hospital, University of Teramo, Piano D’Accio, SP18 64100 Teramo, Italy; 2Department of Conservation and Research, Memphis Zoo, Memphis, TN 38112, USA; 3Faculty of Veterinary Medicine, Avian and Exotic Animal Clinic, University of Veterinary and Pharmaceutical Sciences Brno, Palackeho Tr. 1946/1, 61242 Brno, Czech Republic

**Keywords:** ultrasonography, conservation, follicles, oviduct, anatomy, captive breeding, endangered species, reptile

## Abstract

**Simple Summary:**

A detailed knowledge of reproductive biology is of paramount importance for establishing ex situ conservation programs. This paper describes in detail the follicular morphology, ovarian cycle and reproductive anatomy of the female Louisiana pinesnake (*Pituophis ruthveni*), which is a threatened North American colubrid. Using ultrasound, we described 7 different stages of the follicles and developed markers of progressive follicular maturation for this species, moreover, we dissected two specimens to describe the macroscopic anatomy of the female reproductive system. Furthermore, we identified the correspondence of the sexual activity peak with a particular follicular stage. This data provides parameters for evaluating growth patterns of ovarian follicles in relation to maturation, ovulation, fertility, and reabsorption that can be useful for conservation of this species as well future conservation programs, commercial breeders, and clinicians monitoring reproductive health and breeding of snakes.

**Abstract:**

Accurate monitoring of reproductive activity is necessary for success of captive breeding and recovery of endangered species. Using ultrasonography, we aimed to describe the stages of follicle development of the endangered Louisiana pinesnake (*Pituophis ruthveni*). Ultrasound procedures were performed weekly for 11 females during the 2020 reproductive season by submerging the last half of an unanesthetized female in water and using a 3.0–10.0 MHz linear array transducer placed and moved along the gastrosteges to explore the whole reproductive tract. The presence of follicles, their size, echogenicity, and stage of development was assessed. We observed small, round, anechoic, linearly aligned previtellogenic follicles in the coelom at the beginning of the reproductive season and found that structures dramatically increased in size and shifted in echogenicity as follicles matured and developed before and after ovulation. We classified follicles according to ultrasonographic appearance into 7 different follicle categories: previtellogenic, early vitellogenic, vitellogenic, preovulatory, peri-ovulatory, post ovulatory, and shelled. Using ultrasound, we developed markers of progressive follicular maturation for the Louisiana pinesnake and identified signs of abnormal development and post ovulatory follicle reabsorption. Detailed description of follicular maturation will be useful to improve captive breeding successes, identify mechanisms of reproductive failure, and develop artificial insemination.

## 1. Introduction

Biodiversity is declining at an alarming rate across the globe. Captive breeding programs are one tool in a multilateral approach to the conservation and recovery of imperiled species [1,2]. While captive breeding programs have been established for several species [3], these programs remain difficult to successfully develop [3,4,5]. One of the challenges of ex situ conservation for a focal species is the detailed knowledge of reproductive biology required [6,7].

The Louisiana pinesnake (*Pituophis ruthveni*) is a large, cryptic colubrid listed as threatened under the United States Endangered Species Act (USFWS 2018) and as Endangered by the International Union on the Conservation of Nature (IUCN) Redlist [8]. The range of the Louisiana pinesnake is restricted to the west Gulf Coastal Plain, between eastern Texas and west-central Louisiana and coincides with that of the longleaf pine (*Pinus palustris*) [9,10,11]. Habitat loss, primarily due to logging and fire suppression, is the biggest threat to the Louisiana pinesnake’s survival in the wild, and an extensive ex situ conservation program has been established to recover the species.

The characterization of female reproductive cycles and the identification of biomarkers of female receptivity are necessary to set the timing of pairings for natural breeding and the development of artificial insemination protocols. Although there is a general understanding of the seasonal reproductive cycle of the Louisiana pinesnake, a detailed description of each stage in the reproductive cycle of females is lacking for this species. Based on population movement data, natural breeding of Louisiana pinesnakes in the wild is thought to occur between mid-April and early June [12]. In captivity, mean gestation time is 33.0 to 64.3 days (mean 50.1 ± 2.1 days) and egg incubation time ranges from 60.2 to 86.5 days (mean 72.4 ± 1.6 days), [12]. According to these data, the estimated wild nesting season of the Louisiana pinesnake occurs during mid-May to late August, with hatching occurring in October [12].

Abdominal palpation is the most widely used technique to assess broad- scale reproductive state of snakes [13,14] and has been used to assess the start of follicular activity and diagnose gravid snakes at Louisiana pinesnake breeding facilities for many years. However, fine-scale assessment of follicular morphology and egg development throughout the reproductive cycle requires frequent serial assessments that can be more accurately and safely performed using ultrasonography. Here, we propose to characterize the reproductive cycle of captive Louisiana pinesnakes using ultrasonography [13,14] to describe size and appearance shifts of maturing follicles and developing post ovulated eggs throughout the captive breeding season and use previously described soft tissue echogenicity descriptions of follicle stages for other snakes [13,14,15,16,17,18,19,20,21,22] to develop markers of maturation, ovulation, fertility, and atresia for the Louisiana pinesnake. Louisiana pinesnake oviposits eggs that are 2 times larger than other species previously evaluated throughout the reproductive cycle, therefore, we expected the relatively large size of eggs to drive differences in markers of ovulation and fertility for this oviparous species [14,15]. A detailed description of the different stages of follicular maturation for this species will be useful to establish markers for pair introductions and for the development of optimal timing for artificial insemination.

## 2. Material and Methods

### 2.1. Animals and Husbandry

Throughout the breeding season in 2020, we tracked the ovarian cycle of 12 mature female Louisiana pinesnake (*Pituophis ruthveni*) housed in a Louisiana Pinesnake Breeding facility in Memphis Tennessee, USA. Four individuals were first time breeders and seven were multiparous. Weight of the animals ranged from 1.17 kg to 2.45 kg (mean 1.65 ± 0.48). All procedures and experimentation were performed in accordance with the guidelines of the IACUC committee of the Memphis Zoo (study #2018-1 and 2020-4). At the start of the study, all females were determined to be healthy with no signs of disease or illness upon clinical examination. The females were housed separately in individual Neodesha plastic cages, with newspaper substrate, and provided large hides and free access to water. Females were fed weekly unless otherwise specified. To simulate the natural seasonal cycles of the temperate latitude of in situ populations, we provided a cooling period to simulate brumation for approximately 90 days. This cooling period was performed according to Louisiana pinesnake SSP captive breeding recommendations. Beginning on 1 December 2019, building temperatures were lowered from 28.3 °C to a minimum of 10 °C. This was accomplished by slowly reducing temperatures by up to 10° every 3–5 days until the minimum temperature of 10 °C was reached. Once the minimum temperature was reached, building temperatures were allowed to fluctuate between 8.9 to 15 °C based on external temperatures, with heating and cooling provided as needed to maintain a mean temperature of 10 °C. Temperature reduction was finalized by 15 December, at which point the artificial lights were turned off. Windows along the outside walls of the building were opened to both provide the appropriate photoperiod cycle and improve airflow through the building to encourage natural shifts in temperature throughout the brumation period. On 2 March 2020, the windows were closed, the lights were turned on (12 h:12 h), and the minimum temperature was increased from 10 °C to 18.5 °C. The temperature was increased by up to 10° every 3–5 days until a minimum temperature of 28.3 °C was reached. By 16 March, the temperatures were allowed to fluctuate between 26.7–29.5 °C with a mean temperature of 28 °C, and the photoperiod was set to a 13 h:11 h cycle. Feeding was resumed once temperatures were at least 25.5 °C and the snakes were visibly active. Each adult snake was offered 2–3 small to medium-sized rats on a weekly basis, with feeding amount adjusted based on the prey response of each individual snake. Throughout the breeding season the building was then allowed to continue to heat naturally to 28.3–29.4 °C, and then air conditioned as needed to maintain 28.3 °C.

### 2.2. Ultrasonography

Reproductive state of the females was assessed with a combination of abdominal palpation and ultrasonography. Ultrasound examinations began the first week of April, and were performed at least weekly on each female, avoiding the time around feeding and digestion. Females were examined by ultrasound until exhibiting signs of imminent pre-lay shed. A Chison Digital Color Doppler ultrasound machine (ECO5Vet, L7M-A, Chison Medical Imaging Co. Ltd., Wuxi, Jiangsu, China) was used for the ultrasound examination. The snakes were manually restrained and the caudal third of the animal was sunken in shallow warm water (28–33 °C), to remove any air trapped between the scales impeding correct visualization [15,17]. A 5.3–10.0 MHz linear array transducer set at 10.0 MHz was placed directly upon the gastrosteges of the snakes. Still Images taken of two to four randomly chosen follicles were analyzed using ImageJ [18]; calibrating the measurement tool to 1 cm scale markers provided by the ECOVET system on the ultrasound images. When follicles increased to a size greater than 5 cm, we used a modified scanning technique with video capture to measure the follicle length [19]. The gallbladder was found for a landmark reference for each examination so that the most cranial structures could be identified, then the entire reproductive tract was examined. Follicles were imaged ventrally in both longitudinal and transverse plane. Width (ventral to dorsal) and length (cranial to caudal) measurements for longitudinal and width (ventral to dorsal) and length (right to left) measurements on transverse plane of follicles were recorded for each female, and follicles were assigned to development categories based on soft tissue echogenicity and previous reptile and snake ultrasound findings [13,14,15,16,17,18,19,20,21,22].

### 2.3. Palpation

Abdominal palpation was routinely performed starting the first week of April by a single experienced operator (S.B.R.). Snakes were fasted at least 3 days prior to palpation to prevent regurgitation and avoid food items in the digestive tract while palpating for follicles. Abdominal palpation is a widely used technique to assess reproductive state of snakes [13,14] and has been used to assess start of follicular activity and diagnosis gravid snakes at Louisiana pinesnake breeding facilities for many years. To perform palpation the female was removed from her enclosure and the technician ran two fingers with slight inward pressure along the ventral scales, down the second half of the body. The presence of string-aligned, round, firm objects with discreet margins indicated developing follicles. During the post ovulatory stage, females were considered gravid when follicles felt large, soft, and did not have discreet margins between structures with some females presenting as pear shaped. At that stage, regular palpation was ceased as not to damage developing eggs or the distended oviduct. This brief, noninvasive maneuver was meant to provide information about follicle presence, number, general size, and to compare the results with ultrasound.

### 2.4. Gross Reproductive Anatomy

To verify anatomical information obtained via ultrasound imagery, the reproductive anatomy of two mature female Louisiana pinesnakes, proceeding from a different institution and deceased for natural reasons, were dissected. The coelomic cavity was accessed cutting the skin between the second and third row of scales on the right lateral body wall and dissecting the gastrosteges and abdominal muscles. Ovaries were readily identified and the mesovarium isolated and dissected. Ovaries, oviduct, and cloaca were then isolated. A 3 French urinary catheter was then inserted into the vaginal pouch to identify the path of the oviducts [23]. Finally, the cloaca was inspected manually dilating the proctodeum, and the opening of the vaginal pouches identified.

### 2.5. Statistical Analysis

Sonographic appearance and size (cm) of structures, palpation findings, as well as dates of mating, skin shedding, egg lay date, and finally numbers of fertile and infertile eggs laid were recorded. The size of follicles at each category of follicular development was compared using a Kruskal–Wallis non-parametric statistical test followed by pairwise comparisons using a Dunn’s post hoc test with a Benjamini-Hochberg correction for multiple comparisons [24]. Separate Kruskal–Wallis tests followed by Dunn’s post hoc tests were conducted on measurements of size (width and length in cm) taken at transverse and longitudinal planes. All statistical analyses were performed using program R (version 3.6.0; Vienna, Austria; https://www.R-project.org accessed on 30 September 2022) and alpha was set at 0.05. Summary statistics are presented as means ± standard deviation.

## 3. Results

### 3.1. Ultrasonography

Snakes did not show any sign of distress during the ultrasound examination. Manual restraint proved to be adequate for the procedure, and the water bath provided excellent image quality. The right ovary was echolocated immediately caudal to the gall bladder, while the left ovary was found to caudal and to the right. The follicular structures were then classified according to soft tissue echogenicity into 7 different categories: previtellogenic follicles, early vitellogenic follicles, vitellogenic follicles, preovulatory follicles, Peri-ovulatory follicles, post ovulatory follicles, and shelled eggs (Table 1).

Ultrasound images are presented in Figure 1 in longitudinal plane, and Figure 2 in the transverse plane. Emerging from brumation, all females exhibited previtellogenic follicles, small, round, string-aligned anechoic structures, <1.5 cm in diameter (0.76 ± 0.42 cm along longitudinal plane). Previtellogenic follicles were the most abundant structures identified from 30 March to 15 April (Figure 1A and Figure 2A), depending on when follicles began to develop for each female. Previtellogenic follicles could be visualized between larger, more developed follicles in some females throughout the reproductive cycle.

From 30 March to 12 May, early vitellogenic follicles (Figure 1B and Figure 2B) were visible, these follicles are larger (1.33 ± 0.73 cm along longitudinal plane) than previtellogenic follicles but are still small, round, anechoic to hypoechoic structures with central spot of echogenicity. Vitellogenic follicles developed an oval shape as evident proliferation of the granulosa cells appeared as a thick peripheral hyperechoic layer surrounding the hypoechoic ooplasm, and yolk began to fill the ooplasm giving the follicles a layered appearance (Figure 1C and Figure 2C). In the vitellogenic stage, the follicles began to dispose longitudinally with the long axis positioned rostro-caudally. These follicles were observed from 13 April to 19 May. Preovulatory follicles became evident as yolk continued to fill the structure, the anechoic area migrates toward the periphery of the ooplasm. This stage exhibited uniform, hyperechoic ooplasm, with an elongated anechoic area in the periphery, and were visible from 13 April to 19 May (Figure 1D and Figure 2D).

Follicles continued to elongate, and the anechoic pocket disappeared. We defined these hyperechoic structures with little to no anechoic pocket, and a thick hypoechoic outline occurring between the preovulatory stage and prior to the females having overt lower body swelling, as peri-ovulatory (Figure 1E and Figure 2E), these were visible between 20 April and 26 May. Being a dynamic process, ovulation is difficult to see using ultrasound.

From 4 May to 15 June post ovulatory follicles were present in conjunction with the female exhibiting a swollen lower body. Post ovulatory follicles are larger and started to develop hypoechoic and hyperechoic layers and the hypoechoic rim became thin. Within a few weeks after the follicles entered the oviduct, shelling became visible. Depending on the extent of calcification the shell was visible as a hyperechoic layer deposed around the hyperechoic egg (Figure 3A,B). First visible shelling corresponded to females showing signs of imminent pre-lay shed. Shelled eggs are considered ready to be laid. Females (*n* = 9) laid eggs 7.6 ± 2.2 days after shedding, between 31 May and 2 July. Immediately following oviposition, string-aligned, anechoic previtelogenic follicles were again apparent in all females assessed within 1 week of lay.

The date of reproductive milestones (end of the cooling period (16 March), preovulatory, peri-ovulatory stage, and lay date for each female was recorded) and length of time between milestones are calculated and summarized in Table 2. During the 2020 breeding season, the females (*n* = 9) oviposited 95 days (range 77–107 days) after the cooling period ended on 16 March and females developed preovulatory follicles 48.9 ± 7.9 days and peri-follicles 40.4 ± 8.1 days before laying eggs.

There were no substantial differences in follicular development found between first time breeders and previously bred females. Nine of 12 of the females laid eggs. Together the females produced 40 fertile of 63 eggs laid (63.5% fertile). We also found several instances of abnormal follicles in females. One female exhibited abnormal follicular distribution with follicles clustered instead of string-aligned in coelom during previtellogenic stage (SB 164, Figure 4). These follicles showed signs of atresia by the vitellogenic stage, stopped progressing, and eventually reabsorbed. Follicle measurements from this female were not used in data analyses. We also identified follicles with abnormal features during the post ovulatory stage that we suspected were signs of deterioration, such as uneven edges, no evident shelling and elongation instead of widening during this stage (Figure 5). These eggs were either laid with thin, noncalcified membranes encasing the yolk, or reabsorbed. Reabsorption of post ovulatory follicles was observed for two females where long hyperechoic post ovulatory structures with discreet margins regressed over a few weeks to what appeared to be thick fluid in the oviduct and uterus.

Comparative size of the different stages of the follicles for 11 of 12 females that produced follicles that progressed through maturation is shown in Table 3 and Figure 6. Non-active, previtellogenic follicles had a mean longitudinal plane length of 0.76 ± 0.42 cm. Follicles undergoing active yolk accumulation increased from 2.58 ± 0.90 cm during vitellogenic stage to 6.29 ± 0.89 cm by peri-ovulatory stage. A few weeks before oviposition shelled eggs had a mean length of 8.76 ± 1.94 cm on the longitudinal plane. Using Kruskal–Wallis non-parametric statistical test we found the size of follicles to differ significantly based on follicle stage across all measurements (length and width) and both planes of view (transverse and longitudinal) (Transverse width: H = 283.95, df = 6, *p* < 0.001; Transverse length: H = 309.21, df = 6, *p* < 0.001; Longitudinal width: H = 197.64, df = 6, *p* < 0.001; Longitudinal length: H = 208.65, df = 6, *p* < 0.001). Post hoc comparison tests between developmental stages found most stages differed significantly based on size. However, comparisons of longitudinal measurements showed late stages of follicle development to be similar in size (Figure 6A,B) while transverse measurements found early stages of development were of similar size (Figure 6C,D).

### 3.2. Abdominal Palpation

Snakes were explored once a week by abdominal palpation by a single experienced operator (S.B.R.). No animal showed any signs of distress during the procedure and no snakes were injured. According to ultrasound observations, the stage and size of follicles at first detection by palpation varied by individual and ranged from vitellogenic (1.6–3.4 cm long) to post ovulatory stage (3.4–5 cm long), although presence of follicles was most frequently palpated after the pre-ovulatory stage. Follicle presence was determined for 40% of the females by the preovulatory stage, while detection of follicular presence increased to 80% relative to ultrasonography by periovulatory stage. The capability of determining the stage of the follicles was limited to determining relative size and hardness. An accurate count of follicles was possible during peri-ovulatory stage when follicles were 4–6 cm long and could still be felt as individual, distinct but soft follicles. As follicles continued to develop in the oviduct, the follicles become soft, enlarged to take up the most of the coelomic cavity. Palpation of distinct margins between follicles through the scales was lost at this stage and palpation was ceased. During the post ovulatory stage females became pear shaped and were considered gravid. Regular palpation was ceased as not to damage developing eggs or the distended oviduct.

### 3.3. Gross Anatomy

The female genital system of Louisiana pinesnake repeats the structure already described for other snakes [22,23,25,26]. Vaginal pouches are identified in the dorsal aspect of the urodeum and were easily cannulated in the dead specimen. The two vaginas appear enclosed in a single serosa that runs along the urodeum and coprodeum, afterward, the two oviducts divide and run parallel to the colon, turning upward and sliding ventrally to the surface of the kidney in close contact to the ureters. The right ovary is located cranially to the left, as it has been described in most snakes (Figure 7). The right ovary is in close proximity of the gall bladder, and the left ovary is located just caudal to the right. A number of follicles in different stages of maturation are found in the ovaries during the breeding season, which were quiescent in the two dead specimens dissected, presenting mostly previtellogenic and atretic follicles.

## 4. Discussion

Although the use of ultrasound has become more common to monitor reproductive status by commercial and private snake breeders there are still limited scientific references using ultrasound to detail follicular cycle of oviparous snakes [16,20,21] and no descriptions of follicular cycles of any endangered snake in a wildlife recovery program. This is the first detailed report of the entire reproductive cycle and follicular morphology throughout maturation of any endangered snake and for the Louisiana pinesnake specifically. One of the major challenges in developing captive successful breeding programs and assisted reproductive technologies, such as artificial insemination, is the lack of detailed knowledge of species-specific reproductive biology; which is particularly lacking for endangered reptiles. In this paper we describe in detail the follicular morphology, ovarian cycle, and reproductive anatomy of the female Louisiana pinesnake (*Pituophis ruthveni*) which produces significantly larger eggs than other North American colubrids (Reichling 1990) and previously described oviparous snakes [16,20,21] Follicle descriptions of reptiles have been traditionally divided into two categories: preovulatory (previtollogenic and vitellogenic) and post ovulatory, using season, mating, or fetal development to distinguish between pre and post ovulation development [13,14,16,21]. Here, using ultrasonography, we present a detailed classification of 7 follicular development stages within these categories from previtellogenic to shelled eggs for the Louisiana pinesnake with the aim of improving our captive breeding program by developing markers of fertility. Females had previtellogenic follicles when examined shortly after emerging from the cooling period in March. Females shifted into active vitellogenesis and developed preovulatory follicles 45.5 ± 13.8 days after the end of the cooling period. The follicles expanded rapidly as yolk deposited, increasing 2 to 3 times the length of preovulatory follicles found in ball python or garter snake [14,17]. We observed introduced male snakes most actively breed females during the peri-ovulatory stage when follicles were large (4–7 cm long) and hyperechoic. We confirmed overt breeding behavior and intromission with signs of mating such as dried semen and rough scales around the vent on females (*n* = 4). Females laid 30–40 days after observed mating (*n* = 9).

Post ovulatory follicles during gestation elongated to a mean of 7.46 ± 1.94 cm in length and shelled in conjunction with females exhibiting signs of impending ecdysis. Shelled post ovulatory eggs reached 8.76 ± 1.68 cm in length on longitudinal plane with no fetal development visible. This length exceeds the maximum size reported for any snake species where follicle size has been followed through late gestation [14,16,20,21]. In this study, females laid approximately 7–10 days after shedding and averaged 6.3 ± 1.7 eggs per clutch (*n* = 9). The mean length combined fertile and non-fertile eggs was 8.66 ± 0.99 cm (*n* = 51) at lay, indicating that ultrasound measurements at the end of gestation were predictive of egg size at lay. The mean live hatchling size at emergence from fertile eggs was 47.28 ± 3.8 cm (total length, *n* = 35). The size of laid eggs, hatchlings, and clutch were within ranges previously reported for the Louisiana pinesnake [27]. Louisiana pinesnakes exhibit a maternal investment strategy of producing large eggs and small clutches (4–6 average); however as a nest has yet to be found in the wild for this species, knowledge of reasons for this strategy is limited. It is possible that body condition, previous history, environment [28], or other natural history characteristics [29,30] and prey preference [31] could contribute to this investment strategy, but exact reasons are unknown and beyond the scope of this paper.

Ultrasound examination of ovaries in snakes has been described in the literature [21,22], and it has proven to be the most reliable method for exploring the ovaries in snakes and other Squamata. Digital palpation is a useful approach for some species for determining the presence and to estimate the number of follicles and eggs in snakes [13], however, detailed imaging is necessary to determine the stage of the follicles and their exact number. Furthermore, the ability to visualize these changes as well as accurately count expected eggs prior to oviposition improved our animal care and allowed for immediate intervention when necessary. Follicular malformations are not uncommon in snakes [32], we observed abnormalities in follicle development as well as reabsorption of post ovulatory eggs and egg-binding. The ultrasonographic images and descriptions in this paper may prove useful for clinicians faced with a snake that is displaying reproductive abnormalities and help guide the decision about the necessity of surgical intervention for an egg-bound female compared to a female that is simply reabsorbing.

Detailing the description of the echogenicity and sizing of follicular maturation of the Louisiana pinesnake using ultrasonography is vital knowledge to improve captive breeding efforts, identify mechanisms of reproductive failure, and continue to develop assisted reproductive technologies for this species. The effectiveness of our classifications to predict optimal timing for artificial insemination and natural mating are beyond the objective of this paper; however, we have developed a framework to assess this in future research. It is unknown if like mammals, optimal timing of fertilization for reptiles requires accurate timing of insemination relative to follicular maturation; however, improved outcomes of fertilization and hatch rate have been shown for corn snakes when follicle maturation was used to time artificial insemination [22]. This study is part of an on-going effort to advance the knowledge of the reproductive biology, develop assisted reproductive methods, and improve the reproductive output of the Louisiana pinesnake assurance colony as part of US Fish and Wildlife strategic recovery plan for the species. Improving the understanding of follicular dynamics and reproductive cycles is paramount to furthering efforts for developing effective assisted reproductive technologies in reptiles.

## 5. Conclusions

We described the reproductive cycle of Louisiana pinesnake including follicular maturation, ovulation, and abnormal follicular growth and atresia as well as post-ovulatory egg development through pre-lay shed using ultrasonography. We detailed the related echogenic and size changes of 7 follicular stages throughout the entire reproductive cycle and this data provides parameters for evaluating growth patterns of ovarian follicles in relation to maturation, ovulation, fertility, and reabsorption that can be useful for captive breeders and clinicians monitoring reproductive health and breeding. Furthermore, we provided the first description of follicle size and appearance with this significant increase in follicular size correlated to overt breeding behavior and thereby presenting a potential optimal timing for artificially inseminating females during the reproductive cycle.

## Figures and Tables

**Figure 1 animals-12-02983-f001:**
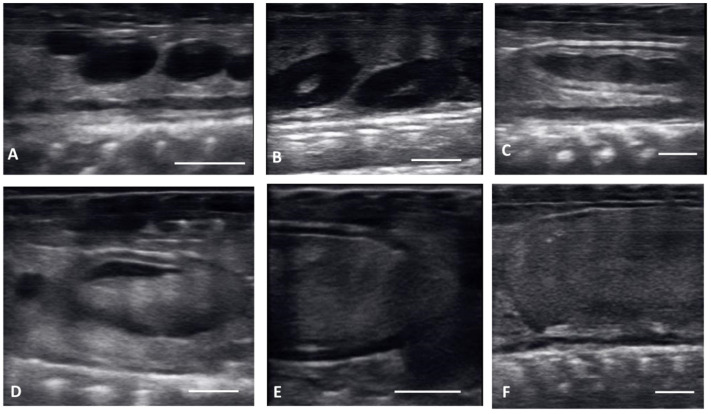
Longitudinal plane of different stages of maturation of the follicles. Clockwise from upper left: (**A**) previtellogenic follicles (PVF): (**B**) Early Vitellogenic (EVF): (**C**) Vitellogenic follicles (VF); (**D**) preovulatory follicle (PrOV); (**E**) Peri-ovulatory follicle (PERI): (**F**) post ovulatory follicles (POV). White bars on picture represents 1 cm.

**Figure 2 animals-12-02983-f002:**
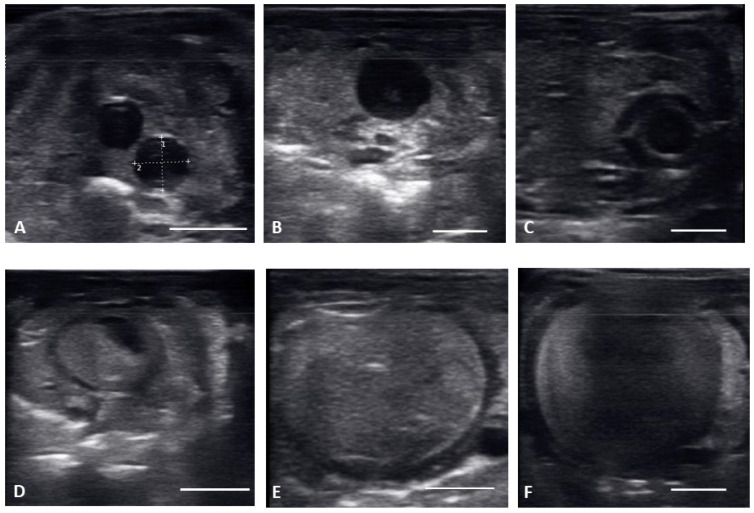
Transverse plane of different stages of maturation of the follicles. Clockwise from upper left: (**A**) previtellogeic follicles (PVF); (**B**) Early Vitellogenic (EVF); (**C**) Vitellogenic follicles (VF); (**D**) preovulatory follicle (PrOV); (**E**) Peri-ovulatory follicle (PERI); (**F**) post ovulatory/gravid (POV). White bars on picture represents 1 cm.

**Figure 3 animals-12-02983-f003:**
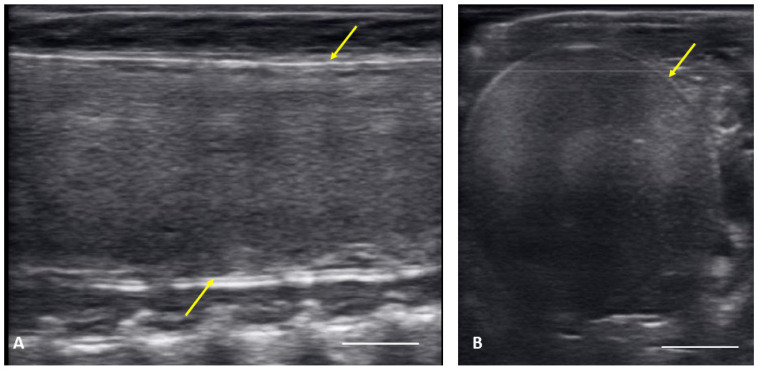
Image of one shelled egg in the oviduct on longitudinal plane (**A**) and transverse plane (**B**), the calcified layer starts to be clearly visible around the structure (yellow arrows), which distinguish it from the post ovulatory follicles.

**Figure 4 animals-12-02983-f004:**
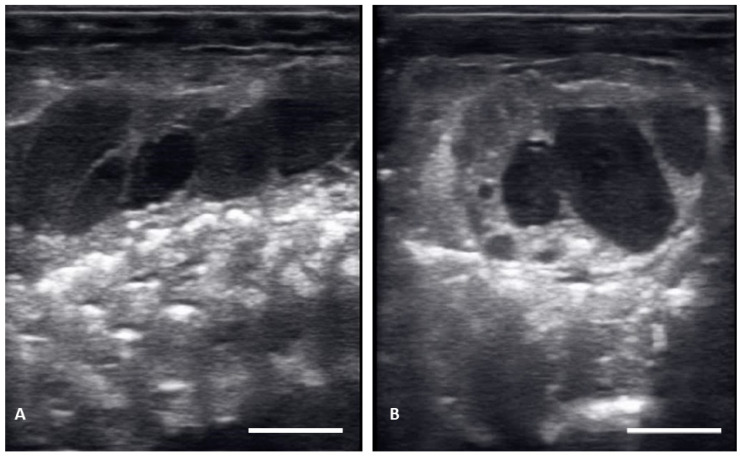
Abnormal distribution of follicles in the ovary on longitudinal plane (**A**) and transverse plane (**B**), These structures remained clustered and did not progress.

**Figure 5 animals-12-02983-f005:**
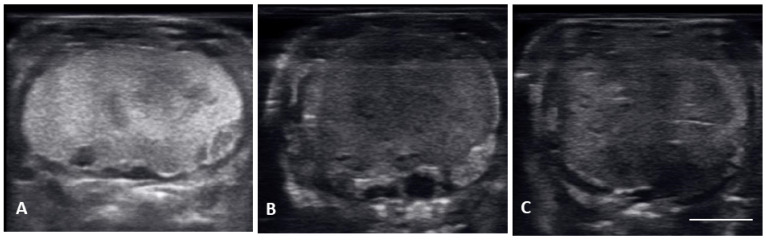
Images of multiple non-fertile post ovulatory follicles in degeneration. All the follicles had uneven outlines and non-uniform, heterogenous echogenicity. (**A**) The entire follicle became hyperechoic, this follicle had to be surgically removed. (**B**) uneven outline and large anechoic areas, this female laid unfertilized slugs. (**C**) thick hyperechoic uneven edges forming with small anechoic pockets dotting the hyperechoic follicle; these follicles were reabsorbed.

**Figure 6 animals-12-02983-f006:**
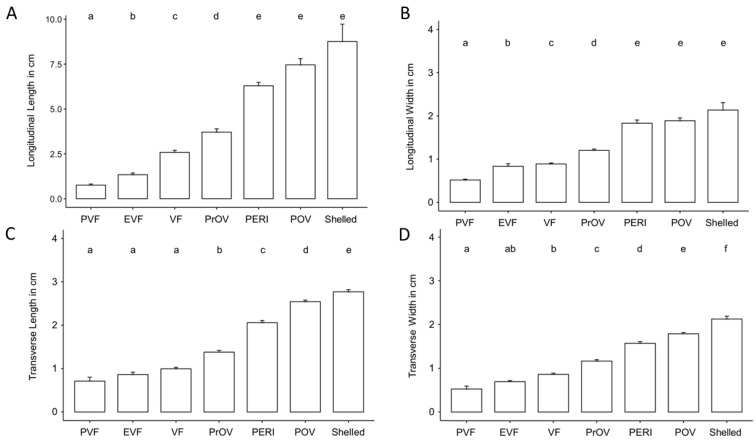
Column graph showing the increase in length and width on longitudinal plane (**A**,**B**) and length and width on transverse plane (**C**,**D**) according to the stage of the follicles for 11 female Louisiana pinesnakes. Abbreviation definitions are in Table 1, bars represent Mean ± s.d. and different letters over the bars indicate statistically different groups as determined by a Kruskal–Wallis non-parametric statistical test followed by a Dunn’s post hoc test with a Benjamini-Hochberg correction for multiple comparisons.

**Figure 7 animals-12-02983-f007:**
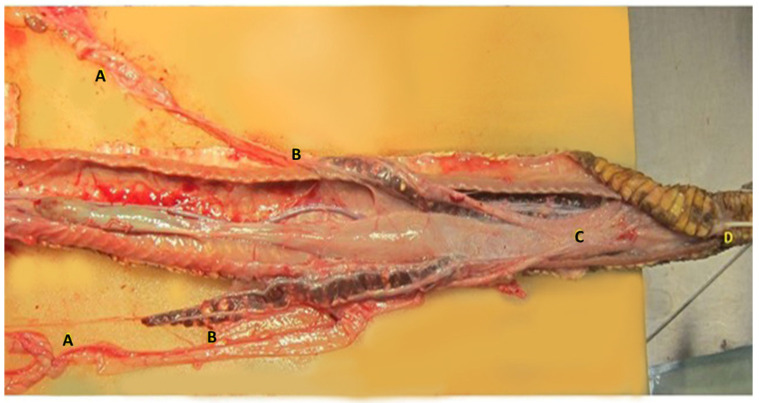
Dissection of the female genital system of the Louisiana pinesnake. The caudal third of the animal is shown. The right ovary is located cranially to the left, as it is described in most snakes (**A**). The two oviducts divide and run parallel to the colon (**C**), turning upward and sliding ventrally to the surface of the kidney in close contact to the ureters (**B**). The two vaginas appear enclosed in a single serosa that runs upon the urodeum and coprodeum, in the right vagina a 3 french tom cat catheter was inserted (**D**) cloaca.

**Table 1 animals-12-02983-t001:** Categories of follicle development stages.

Previtellogenic Follicles (PVF)	Anechoic Follicles, Small and Rounded, String-Aligned
**Early vitellogenic follicles (EVF)**	Small, round follicles showing hyperechoic outline around an anechoic middle and hyperechoic center.
**Vitellogenic follicles (VF)**	Oval shaped follicles showing a thickening hyperechoic outline as granulosa cell layer increase, hyperechoic ooplasm, and an anechoic core centrally located.
**Preovulatory follicles (PrOV)**	Elongating and increasing hyperechoic ooplasm as follicles are filled with yolk, with an elongated anechoic area located in the periphery of the ooplasm.
**Peri-ovulatory follicles (PERI)**	Elongated follicles, mostly homogenous hyperechoic structure with thick hypoechoic outline. Before female is overtly swollen.
**Post ovulatory follicles (POV)**	Ovulated follicles, passed into the oviduct showing an hypoechoic surrounding, but no visible calcified layer.
**Shelled eggs (Shell)**	Ovulated follicles, passed into the oviduct and surrounded by a calcified layer.

**Table 2 animals-12-02983-t002:** Summary table of mean length of time in days from seasonal reproductive milestones for each female to lay date for 9 female Louisiana pinesnakes that laid fertile and unfertile eggs in 2020. CP = cooling period, which ended on 16 March. s.d. = standard deviation.

	End CP to PrOV	End CP to PERI	End CP to Lay	PreOv to Lay	PERI to Lay	Lay Shed to Lay
**Days**	45.5	53.6	95.1	48.9	40.4	7.6
**s.d.**	13.8	13.1	10.8	7.9	8.1	2.3

**Table 3 animals-12-02983-t003:** Summary table of size (length and width in cm) and timing of development for follicles at seven categories of development observed in 11 female Louisiana pinesnakes using ultrasonography in the transversal and longitudinal plane.

	PVF	EVF	VF	PrOV	PERI	POV	SHELL
**Date(s)**	30 March to 7 May 2020	30 March to 7 May 2020	7 April to 12 May 2020	13 April to 19 May 2020	20 April to 26 May 2020	11 May to 15 June 2020	11 May to 22 June 2020
**Longitudinal Plane (cm) (±s.d.)**							
**Mean Length**	0.76 ± 0.42	1.33 ± 0.73	2.58 ± 0.90	3.71 ± 1.2	6.29 ± 0.89	7.46 ± 1.94	8.76 ± 1.94
**Min**	0.39	0.47	0.74	1.62	3.92	3.8	6.5
**Max**	2.34	3.51	4.85	4.85	8.27	11.57	11
**Mean Width**	0.52 ± 0.13	0.84 ± 0.43	0.89 ± 0.17	1.20 ± 0.21	1.83 ± 0.36	1.89 ± 0.38	2.14 ± 0.42
**Min**	0.32	0.425	0.517	0.79	1.34	0.932	1.63
**Max**	0.83	2.46	1.31	1.7	2.388	2.83	2.64
**Transverse Plane (cm) (±s.d.)**							
**Mean Length**	0.71 ± 0.42	0.87 ± 0.27	1.0 ± 0.23	1.38 ± 0.27	2.06 ± 0.43	2.54 ± 0.45	2.77 ± 0.38
**Min**	0.21	0.57	0.39	0.96	1.17	1.26	1.63
**Max**	2.07	1.67	1.43	1.88	3.438	3.49	3.54
**Mean Width**	0.53 ± 0.30	0.69 ± 0.13	0.86 ± 0.19	1.16 ± 0.23	1.57 ± 0.33	1.79 ± 0.35	2.13 ± 0.47
**Min**	0.17	0.49	0.26	0.56	0.98	0.93	1.16
**Max**	1.5	1.09	1.4	1.71	2.47	2.53	3.33

## Data Availability

All associated data are publicly available from the authors or can be accessed at the Figshare archive https://10.6084/m9.figshare.21430527 accessed on 30 September 2022.

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
