# Peer review of "Ultrasound Description of Follicular Development in the Louisiana Pinesnake (Pituophis ruthveni, Stull 1929)"

_animals, 2022, doi:10.3390/ani12212983_

Round 1

Reviewer 1 Report

Good paper. Just revise a handful of items in the annotated manuscript

Author Response

Dear Reviewer 1,

Thank you for your comments and suggestions. The comments were loaded into word document according to the reviewers comments placed in a PDF throughout the paper by co-author Melanie Richter (MMR) and we responded accordingly within the paper. 

Most of the comments were for suggestion of references or wording and we thank you for noticing these missing and we have adjusted the manuscript accordingly. 

Please see the word file attached to view the comments and adjustments. 

The reviewer did leave one comment that I have added here and responded below in red.  

Reviewer 1: I recommend a brief commentaire here on the evolutionary meaning of large offsprings in reptiles, with permit the handling of their first, and rather large, prey; reference:

Congdon, J. D. (1989). Proximate and evolutionary

constraints on energy relations of reptiles.

Physiological Zoology 62, 356-373.

would be suitable here

Response:

Thank you for the suggested topic and reference, we did not find the literature to provide insight for LPS hatchling size, instead  I added some natural history about the LPS to adds some insight on why a larger hatchling may be adaptive.

Reviewer 2 Report

Dear Editor,

The work is interesting and worthy of being published in Animals journal. However it could be improved in some parts.

-Line 21-22: abdominal palpation should not be considered a reliable method for follicle identification, especially in medium and small-sized snakes. Also, this technique is operator dependent and it is also dangerous if performed with too much compression. Authors should only mention the technique with less emphasis, reporting the potential risk of follicol rupture.

-Line 38-40: What do you mean for breeding effort? Moreover, the authors should  mention of the "best" period in which the male could be moved together with female improving the mating chance.

-Line 51: authors should provide the usefulness of this work in the clinical practice and also for commercial purposes, not only for the specie conservation.

-Line 376: A recent and similar work was lead on Boa constrictor: this should be cited in the text: Monitoring of the reproductive cycle in captive-bred female Boa constrictor.

-Table 1: please check the correct dimension of previtellogenic follicles (<15 cm seems too much)

-Picture 1: please remove the background, the blood and tissue fragment. The picture is not adequate for publication as presented.

Author Response

Thank you to reviewer two for comments and suggestions we have placed the reviewer comments below and have answered in red. For comments that required changes to the manuscript we added the comments to the manuscript and then adjusted accordingly. These changes can be found directly in manuscript and we have also addressed them below. 

We thank you for your time and efforts and we feel the manuscript has thus been improved.

Reviewer 2: “-Line 21-22: abdominal palpation should not be considered a reliable method for follicle identification, especially in medium and small-sized snakes. Also, this technique is operator dependent and it is also dangerous if performed with too much compression. Authors should only mention the technique with less emphasis, reporting the potential risk of follicle rupture.”

Response: The author's do not agree that this technique (abdominal palpation) poses significant risk to LPS. It is a very common practice and a literature search found no supporting evidence for the risk of follicle rupture in snakes (except of retained ova that have become friable). Although there does appear to be more need for caution in lizard species. We will make it clear that no palpations were conducted after the animal showed a distended coelom, corresponding to the presence of very large follicles. In 3.2 we specify: “Snakes were regularly explored by abdominal palpation by a single experienced operator (S.R.). No animal showed any signs of distress during the procedure and no snakes were injured. … Regular palpation was ceased at post ovulatory stage as not to damage developing eggs.”

Verbiage added to the methods section: “Palpations were only performed by individuals with extensive experience and were not conducted on animals with enlarged coelom, corresponding to the presence of very large follicles.” As ultrasound ultimately gives more reliable and accurate information and allows for quantification of follicle size and see the internal structures of the follicle we did reduce the emphasis on abdominal palpation and mentioned that ultrasound allows for safer serial examinations.

Reviewer 2: -Line 38-40: What do you mean for breeding effort? Moreover, the authors should  mention of the "best" period in which the male could be moved together with female improving the mating chance.

Response: We specify in the discussion that defining an "optimal period for breeding" is outside of the scope of this work. I changed the wording from "efforts" to "successes" as that is what we mean by 'improved breeding efforts'. Also - line 298: "We observed introduced male snakes most actively breed females during the peri-ovulatory stage when follicles were large (4-7 cm long) and hyperechoic and females laid 30-40 days after observed mating. "

Reviewer 2: Line 51: authors should provide the usefulness of this work in the clinical practice and also for commercial purposes, not only for the specie conservation.

Response: Line 317 added text: The ultrasonographic images and descriptions in this paper may prove useful for clinicians faced with a snake that is displaying reproductive abnormalities and help guide the decision about the necessity of surgical intervention for an egg-bound female compared to one that is reabsorbing.

Reviewer 2: Line 376: A recent and similar work was lead on Boa constrictor: this should be cited in the text: Monitoring of the reproductive cycle in captive-bred female Boa constrictor.

Response: This citation has been added. Thank you for bringing this omission to our attention.